# On the accuracy of aerosol photoacoustic spectrometer calibrations using absorption by ozone

Nicholas W. Davies[1,2], Michael I. Cotterell[1,2], Cathryn Fox[2], Kate Szpek[2], Jim M. Haywood[1,3] and Justin M. Langridge[2]

[1]College of Engineering, Mathematics and Physical Sciences, University of Exeter, Exeter, EX4 4QF, United Kingdom
[2]Observation Based Research, Met Office, Exeter, EX1 3PB, United Kingdom
[3]Earth System and Mitigation Science, Met Office Hadley Centre, Exeter, EX1 3PB, United Kingdom

*Correspondence to*: Justin M. Langridge (justin.langridge@metoffice.gov.uk)

**Abstract.** In recent years, photoacoustic spectroscopy has emerged as an invaluable tool for the accurate measurement of light absorption by atmospheric aerosol. Photoacoustic instruments require calibration, which can be achieved by measuring the photoacoustic signal generated by known quantities of gaseous ozone. Recent work has questioned the validity of this approach at short visible wavelengths (404 nm), indicating systematic calibration errors of the order of a factor of two. We revisit this result and test the validity of the ozone calibration method using a suite of multi-pass photoacoustic cells operating at wavelengths 405, 514 and 658 nm. Using aerosolised nigrosin with mobility-selected diameters in the range 250–425 nm, we demonstrate excellent agreement between measured and modelled ensemble absorption cross sections at all wavelengths, thus demonstrating the validity of the ozone-based calibration method for aerosol photoacoustic spectroscopy at visible wavelengths.

## 1 Introduction

Uncertainty in the radiative forcing that drives climate change is dominated by the poorly constrained impact of aerosols on Earth's radiation budget, with aerosol-radiation interactions contributing a global mean effective radiative forcing of -0.45 (-0.95 to +0.05) W m$^{-2}$ (Myhre et al., 2013; Stocker et al., 2013). Aerosol single scattering albedo (the ratio of scattering efficiency to total extinction efficiency) is one of the key inputs used in radiative transfer models to represent aerosol optical behaviour and is amongst the largest contributors to uncertainty in direct radiative forcing (McComiskey et al., 2008). The accuracy of the single scattering albedo is limited by knowledge of aerosol absorption properties (Boucher et al., 2013; Stier et al., 2013). Black carbon, a carbonaceous material formed by incomplete combustion, absorbs strongly at visible wavelengths and has been shown to have significant climate implications (e.g. Bond et al., 2013; Stocker et al., 2013). Evidence also shows that light-absorbing organic aerosols, commonly referred to as brown carbon, absorb strongly towards wavelengths in the ultra-violet (Andreae and Gelencsér, 2006). The optical properties of black and brown carbon are poorly captured in climate models owing in part to a lack of detailed measurements (Alexander et al., 2008; Bond et al., 2013; Cappa et al., 2012; Lack and Cappa, 2010; Liu et al., 2015a, 2015b; Myhre et al., 2013; Saleh et al., 2013; Wang et al.,

2016).

Traditionally, aerosol absorption coefficients are retrieved via the difference method or by using filter-based absorption photometry. The difference method involves subtracting separate measurements of the scattering coefficient from the extinction coefficient, leading to large uncertainties in the calculated absorption arising from the quadrature combination of errors in the two measurements. These uncertainties in absorption are particularly high at single scattering albedos corresponding to weakly absorbing aerosols (Bond et al., 1999; Lack et al., 2006; Strawa et al., 2003). Filter-based absorption measurements rely on determining the change in light transmittance across a particle-laden filter. This method is subject to biases and, although a range of correction schemes have been proposed (Arnott et al., 2005; Bond et al., 1999; Coen et al., 2010; Müller et al., 2014; Schmid et al., 2006; Virkkula et al., 2005; Virkkula, 2010; Weingartner et al., 2003), aerosol absorption biases in the range of 20–200 % can remain (Backman et al., 2014; Cappa et al., 2008; Lack et al., 2008; Müller et al., 2014). Photoacoustic spectroscopy (PAS) is a state-of-the-art technique that measures absorption directly for particles in their natural suspended state (Arnott et al., 1999). It can be used to differentiate between the absorption enhancement due to the lensing effect of coated black carbon and the absorption contribution due to brown carbon by utilising thermally denuded channels (Lack et al., 2012). For these reasons, it has become the technique of choice for measuring aerosol absorption.

The PAS principle relies on converting energy from a light source into sound. Light-absorbing media, such as aerosol, transfer intensity-modulated electromagnetic energy into thermal energy that heats the surrounding air. This gaseous heating generates a pressure wave that propagates radially away from the heated aerosol particle. The periodic heating driven by the modulated light beam is performed at the same frequency as a standing acoustic (pressure) wave eigenmode of the photoacoustic cell. Excitation of a PAS cell eigenmode over repeated heating cycles amplifies the photoacoustic pressure signal for detection by a microphone located within the PAS cell (Arnott et al., 1999; Miklós et al., 2001; Moosmüller et al., 2009). The amplitude of the microphone signal at the modulation frequency is related to the sample absorption coefficient through calibration. Multi-pass optics are commonly used to increase the circulating light intensity within the PAS cell, which provides increased sensitivity through increased sample heating. This approach is advantageous for aerosol studies compared to single pass methods employing higher laser powers, as it increases sampling heating without exposing individual particles to large temperature changes which could lead to loss of semi-volatile species through evaporation (Lack et al., 2006; McManus et al., 1995). Biases associated with PAS include a lack of proportionality between the photoacoustic signal and the aerosol absorption cross section for particles with radii greater than 0.7 µm (Cremer et al., 2017). This is not an issue for the current study, which uses an impactor to remove particles with radii > 0.5 µm; see Sect. 2.4.

There exist a number of options for calibrating photoacoustic spectrometers including use of nitrogen dioxide (Arnott et al., 2000; Nakayama et al., 2015), polydisperse kerosene soot (Nakayama et al., 2015), oxygen (Gillis et al., 2010; Tian et al.,

2005) and ozone (Bluvshtein et al., 2017; Lack et al., 2006; Lack et al., 2012). Ozone was chosen as the calibrant for our PAS cells, in part as nitrogen dioxide has been shown to introduce uncertainty in calibrations at 405 nm due to photolysis (Lack et al., 2012) and generation of aerosol particles is challenging in the field. Gaseous ozone has been used successfully to calibrate photoacoustic spectrometers operating at laser wavelengths of 532 nm, with reported absorption accuracies of 1–

5 % (Lack et al., 2006; Lack et al., 2012). Demonstration of the validity of the ozone calibration approach involved the comparison of PAS measurements to model absorption calculations for laboratory-generated absorbing particles, such as nigrosin dye. Recently, Bluvshtein et al. (2017) performed similar experiments to probe the validity of the ozone calibration approach at 404 nm. They found a factor ~2 discrepancy between the PAS response to ozone and nigrosin, which was attributed to an unspecified issue with ozone measurements at these wavelengths. This result has significant implications for

photoacoustic spectrometer ozone calibrations at short-visible wavelengths, suggesting that they would lead to overestimation of aerosol absorption by a factor ~2. The focus of this study is to re-evaluate this result.

Given the importance of the Bluvshtein et al. (2017) work in motivating this study, we provide a brief overview of the experiments here. Ozone was generated from high purity (99.999 %) oxygen using either a corona discharge ozone generator

or, for lower concentrations, a UV lamp. Ozone concentrations in the range 250–1900 ppm were generated and diluted with dry nitrogen in the ratio 1:10 $O_2$-$O_3$:$N_2$. Ozone absorption coefficients were measured using a cavity ring-down spectrometer (CRDS). CRDS measurements made in series versus parallel to the PAS detection cell indicated little difference in ozone concentration and thus minimal loss through the PAS system. Nigrosin was atomised from solution, dried to <10 % relative humidity, size-selected using a differential mobility analyser (DMA) to yield mobility diameters in the range 250–325 nm

and passed through an impactor to remove multiply charged particles. The aerosol stream flowed through a PAS cell at 0.6 L min$^{-1}$, which was then split evenly between the CRDS cell and a condensation particle counter (CPC). Particle concentrations in the range 200–1500 cm$^{-3}$ were used. PAS-measured ensemble aerosol absorption coefficients were compared to modelled values computed using Mie-Debye-Lorenz theory (hereafter referred to as *Mie* theory) in combination with the size-selected particle diameters passed by the DMA and complex refractive indices determined via spectroscopic ellipsometry.

Experiments were repeated for Pahokee peat fulvic acid and Suwannee river fulvic acid aerosol. The discrepancy between the PAS-measured ensemble absorption coefficients and absorption coefficients calculated using Mie theory differed by a factor of two for all three test aerosols. Several suggestions for the discrepancy were provided, including contamination by $NO_2$ and generation of light-scattering particles due to reaction of $O_3$ with the walls of the instrument. However, no evidence supporting these theories was provided. PAS measurements at wavelengths other than 404 nm were also not available, which

prevented an independent check of the PAS responses to ozone and nigrosin at wavelengths that have been reported previously to be well-calibrated using the ozone approach (e.g. Lack et al., 2006).

In this study, we use a suite of multi-wavelength PAS and CRDS measurements to evaluate the suitability of ozone as a PAS calibrant gas. We follow the method described above whereby PAS-measured ensemble absorption cross sections for

aerosolised nigrosin are compared to model calculations. Importantly, this comparison is evaluated for three visible wavelengths including the 405 nm wavelength pertinent to the work of Bluvshtein et al. (2017). In the following section, we provide a description of the experimental setup including the photoacoustic and cavity ring-down spectrometers, ozone calibration apparatus, aerosol generation system and the method by which modelled ensemble aerosol absorption cross sections were calculated. Experimental results and discussion are presented in Sect. 3.

## 2 Methodology

### 2.1 Photoacoustic spectrometer

Photoacoustic detection cells based on the dual-resonator design of Lack et al. (2012) were used. These cells were identical to those used by Bluvshtein et al. (2017) except that the planar windows were replaced with Brewster angle windows (Thorlabs, BW2502), which minimised reflection losses within the multi-pass optical system. Each PAS cell consisted of two cylindrical resonator cavities (a lower *signal* and upper *reference* resonator) coupled through buffer volumes on either side for noise suppression (Lack et al., 2006). The cells were manufactured from aluminium and had a total volume of 200 $cm^3$. Individual resonant cavities had dimensions of 110 mm length and 9.5 mm radius. The sample passed through both resonators but laser light passed only through the lower, *signal* resonator. The cell was positioned within a multi-pass optical system formed by two cylindrical mirrors located outside of the PAS cell with mirror radii of curvature of 430 mm (front mirror, closest to laser) and 470 mm (back mirror). The concavities of the two mirrors were rotated 90° to each other. Each mirror was coated with a wavelength-specific dielectric coating to yield reflectivities exceeding 99.9 %. Toptica iBeam Smart (Toptica Photonics) lasers with wavelengths 405, 514 and 658 nm generated light with powers 300, 100 and 130 mW respectively. Laser wavelengths and line widths were measured using an Avantes spectrometer (CompactLine) for the blue and green wavelengths and a Hamamatsu spectrometer (C11697MB) for red wavelengths. Light was injected into each multi-pass system through a 2 mm hole in the centre of the first mirror. Light exiting the multi-pass system was measured using a photodiode (Thorlabs, S121C) positioned behind the second mirror. In an optimally aligned system, the laser would pass through the acoustic resonator 182 times (Lack et al., 2012; Silver et al., 2005). However, no effort was made to achieve this limit in the current system. Alignment was conducted by visual inspection of the spot pattern only, which almost certainly resulted in a lower number of passes. Quantifying the number of passes through the resonator was not critical. Light exiting the resonator was measured using a photodiode, which allowed the PAS signal to be corrected for any laser power or alignment instability (Lack et al., 2012). The acoustic signal was detected using microphones (Knowles Acoustics, EK-23132) positioned half way along the lengths of each resonator to coincide with the pressure antinode corresponding to the lowest-order ($n$ = 1) acoustic eigenmode of the photoacoustic cell. The responses from the two microphones were passed through a differential amplifier and Fourier-transformed to the frequency domain. The photoacoustic response is defined as the magnitude of the frequency domain response at the $n$ = 1 eigenfrequency and is referred to hereafter as the *integrated area* (IA). A speaker (Knowles Acoustics, ES-23127-000) was located in the reference resonator to enable periodic

measurement of the cell resonant frequency and quality factor (Lack et al., 2012). Following each speaker measurement, the laser modulation frequency was automatically adjusted to match the derived cell resonance frequency. Section 2.3 provides details regarding PAS corrections for cell resonance properties and laser power. Aerosol absorption coefficients ($Mm^{-1}$) measured by the photoacoustic spectrometers were converted to ensemble absorption cross sections ($m^2$) for comparison to theoretical calculations by dividing by the aerosol number concentrations reported by a CPC (see Sect. 2.4). The ensemble absorption cross section represents the mean of the absorption cross sections corresponding to a range of particles sizes, for example from multiply charged particles (see Sect. 2.5). The ensemble absorption cross section is hereafter referred to as the absorption cross section.

## 2.2 Cavity ring-down spectrometer

Cavity ring-down spectroscopy is a highly sensitive technique used for measuring the optical extinction coefficient of gases and particulate matter (O'Keefe and Deacon, 1988; Romanini et al., 1997). The CRDS system employed in this study was similar to that in Langridge et al. (2011) and only the differences will be highlighted here. All lasers (Toptica, iBeam Smart-S) were continuous wave diode lasers, operated with square wave modulation at a frequency of 2000 Hz. Lasers were protected from back-reflections using Faraday isolators (Thorlabs, IO-5-405-LP and IO-3D-660-VLP). A 658 nm laser (130 mW) pumped the red cell and a 405 nm laser (300 mW) pumped the blue cell. The laser spectral widths were > 100 GHz and much larger than the free spectral range of the optical cavities (~ 350–400 MHz). This allowed passive coupling to occur rather than relying on an active mechanism to match the laser frequency to a cavity mode. The CRDS cells were made out of aluminium. Cavity mirrors were manufactured from fused silica with wavelength-specific coatings, 25 mm diameter, 1 m radii of curvature and reflectivities in excess of 99.99 % (Layertec GmbH, red 660 nm; CVI Laser Optics, blue 405 nm). A high purity zero-air flow (BOC, 270028-L) set to 10 $cm^3$ $min^{-1}$ per mirror was introduced across the mirrors to prevent contamination. Photomultiplier tubes detected light exiting the cavity (Hamamatsu, H9433-201) and were protected from stray light using narrow band interference filters (Thorlabs, FB405-10 and FB660-10). Each time the laser turned off, the cavity output signal decayed exponentially. The signal was fitted to a single exponential function to extract the 1/e folding time, otherwise known as the cavity ring-down time. The extinction coefficient, $\alpha_{ext}$, ($Mm^{-1}$) was calculated using

$$\alpha_{ext} = \frac{R_L}{c}\left(\frac{1}{\tau} - \frac{1}{\tau_0}\right),\tag{1}$$

where $R_L$ is the ratio of the physical length of the cavity to the length over which sample was present, $c$ is the speed of light and $\tau$ and $\tau_0$ are the ring-down times for a cavity with and without scattering/absorbing species. The $\tau_0$ times for both the 405 and 658 nm CRDS channels used in this study were measured before and after experiments where aerosol was passed through the optical cavities. These $\tau_0$ varied over time by only a small amount due to changes in cavity alignment, cleanliness and the sample pressure. However, typical representative times were 23.1 μs (405 nm) and 34.2 μs (658 nm). Cavity mirror-to-mirror lengths ranged from 371–423 mm yielding geometric $R_L$ factors in the range 1.150–1.173. The $R_L$

factor appropriate for aerosol measurements was determined from the geometric dimensions of the detection cell. As highlighted by Fuchs et al. (2008), the $R_L$ factor for detection of gaseous species can be different from this value, due to the ability of gaseous samples to diffuse. We determined the gaseous $R_L$ factors by measuring the change in the ring-down times for filtered air plus ozone in (i) standard operation whereby ozone partially diffuses into the volume between the sample inlet and mirror and (ii) non-standard operation whereby ozone was fully mixed into the volume between the sample inlet and mirror by pulling the ozone-laden air out of the cavity through the mirror purge lines. This resulted in $R_L$ factors 1.05 (658 nm) and 1.04 (405 nm). The CRDS extinction measurement accuracy was evaluated by Langridge et al. (2011) to be better than 2 %. Extinction coefficients were converted to ensemble extinction cross sections ($m^2$) by dividing by the aerosol number concentrations measured using a CPC (see Sect. 2.4). The ensemble extinction cross section is hereafter referred to as the extinction cross section.

## 2.3 Ozone calibration

Gaseous ozone was generated using a coronal discharge ozone generator (Longevity Resources, EXT120-T) from high purity oxygen (99.999 %, BOC, grade N5.0). The ozone-laden stream was split approximately evenly between the PAS and CRDS cells using a manifold equipped with 300 μm diameter orifices, as shown in Fig. 1. Teflon tubing was used throughout the flow system to minimise contamination and to reduce ozone losses.

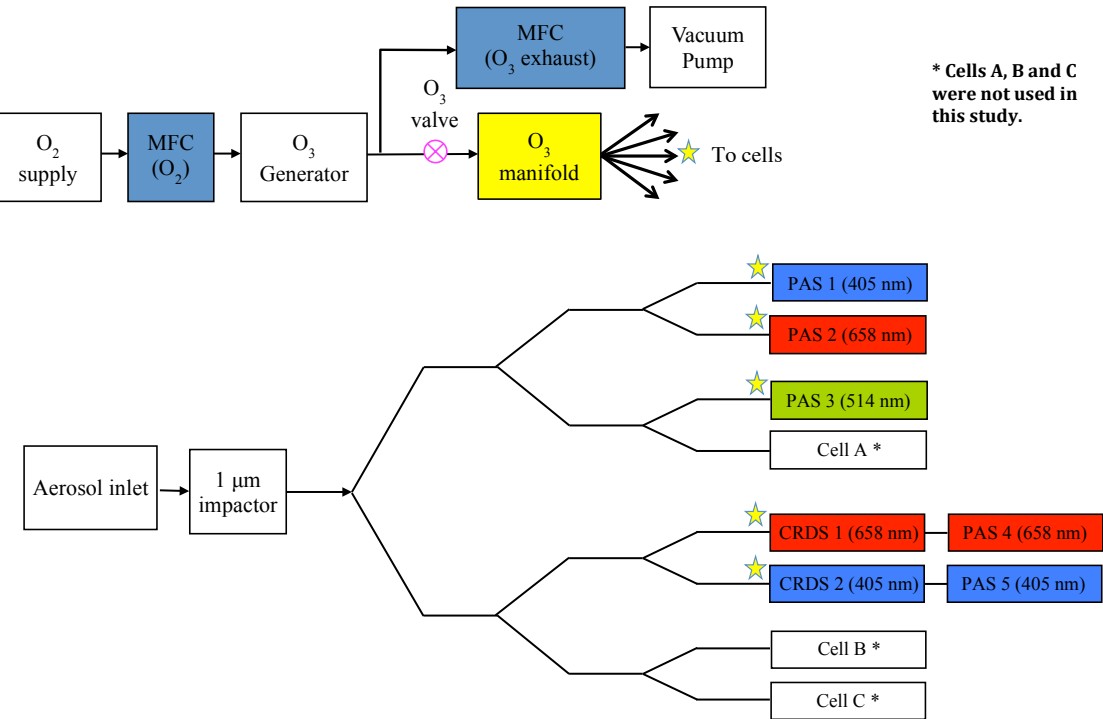

**Figure 1: Schematic diagram of the PAS/CRDS experimental setup including the ozone generation system and the relative positions of the PAS and CRDS cells. The stars indicate the ozone flow path, which entered the cells through different ports to the main aerosol flow. The PAS/CRDS wavelengths are centred at 405, 514 and 658 nm respectively. Abbreviations: 'MFC': mass flow controller.**

The 405 nm and 658 nm CRDS cells quantified ozone concentrations for calibration of all five PAS cells. For PAS cells in series with the CRDS channels (PAS 4 and PAS 5), the CRDS-measured extinction coefficients were used directly to calibrate the corresponding in-line PAS channel measurements of IA. This calibration relation between sample extinction and IA is quantified at multiple values of ozone concentration, controlled by varying the discharge frequency on the coronal ozone generator. For PAS cells operated in parallel, it was necessary to measure accurately the relative ozone splitting ratio with respect to the CRDS flow paths. The following section details the method for characterising this ratio, which was based on monitoring the resonant frequency shift induced by changing the gas composition, and hence speed of sound within the photoacoustic cells (Miklós et al., 2001).

At the start of each calibration cycle, pure oxygen was introduced into the PAS cells through the ozone manifold at a flow rate of 0.02 L min$^{-1}$ per cell, in addition to the 0.98 L min$^{-1}$ filtered-air flow. Air was filtered using a particle filter (Headline Filters, DIF-LK40). The oxygen displaced a fraction of the filtered-air flow through each cell, changing the gas composition, speed of sound and thus cell resonant frequency, as shown in Fig. 2. The ozone flow splitting, $\Delta v$, between the two 658 nm PAS cells (PAS 2 and PAS 4 in Fig. 1) was calculated using

$$\Delta v = \frac{\left(F_r^{air} - F_r^{air+o2}\right)_{PAS\,4}}{\left(F_r^{air} - F_r^{air+o2}\right)_{PAS\,2}}, \tag{2}$$

where $F_r^{air}$ and $F_r^{air+o2}$ are the PAS cell resonant frequencies of filtered ambient air and filtered ambient air plus oxygen, respectively, during the two highlighted periods in Fig. 2. Similarly, the ozone flow splitting between the 405 nm PAS cells was calculated using PAS 1 and PAS 5. The 514 nm PAS cell was calibrated using the 658 nm CRDS cell, and hence the ozone splitting ratio between PAS cells 3 and 4 was used. The ozone splitting ratio represents the fractional difference in the ozone concentrations within two PAS cells due to unequal flow splitting within the ozone manifold. The ozone splitting ratios, and therefore the ozone-laden flow rates, between two PAS cells located in parallel (for example, the PAS 2 and PAS 4 cells) were in the range 2–28 %. Measuring the ozone splitting between PAS cells using the resonant shift method compared extremely well to in-line mass flow measurements. The 1σ variability between ozone splitting corrections for eight repeat ozone calibrations was ±1.3 %. A summary of the ozone splitting corrections can be found in the supplementary material.

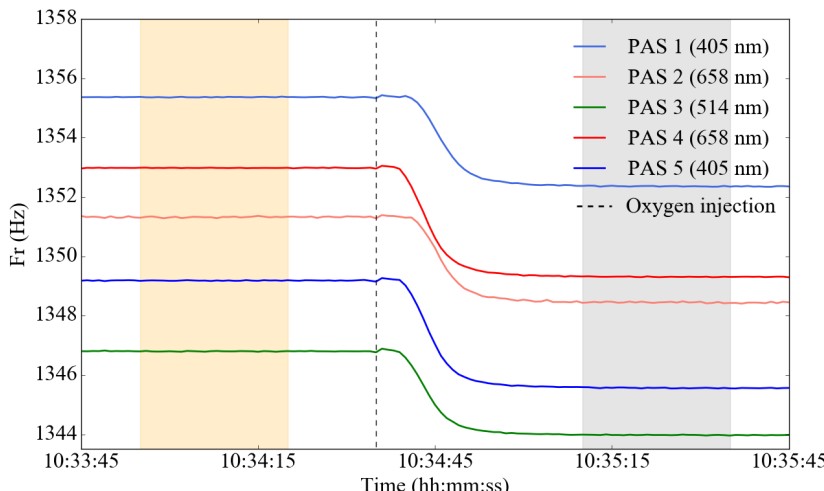

**Figure 2: PAS cell resonant frequency as a function of time. Oxygen was introduced into the filtered ambient air flow at 10:34:35 (dotted line). Mean cell resonant frequencies before (orange highlighted region) and after (grey highlighted region) introducing oxygen were computed during the highlighted times.**

Following measurement of ozone splitting ratios, the ozone generator was powered and the main calibration started. Calibrations involved the stepwise measurement of nine ozone concentration levels, where Fig. 3 shows the PAS and CRDS

responses to ozone at 405 nm. At each ozone level, cell resonant frequencies and quality factors were quantified using the cell speakers. Subsequently, 90 seconds of data were collected from which the mean and standard deviation of PAS IA and CRDS extinction were calculated. Using the minimum and maximum extinction coefficients for ozone in Fig. 3 (1.3 and 27.1 Mm$^{-1}$, respectively), an ozone absorption cross section of $1.62\times10^{-23}$ cm$^2$ at the corresponding CRDS wavelength (405.03 nm) and assuming $2.46\times10^{25}$ molecules of air per cubic metre at the 405 nm CRDS cell temperature and pressure of 21.82 °C and 1001 mb, the ozone concentrations were in the range 33–680 ppmv (Serdyuchenko et al., 2014). Approximately the same levels of ozone were used in all cells. The ratios of ozone extinction coefficients measured in the 405 and 658 nm CRDS cells compared well to the ratio of the literature ozone absorption cross sections. After accounting for uneven ozone flow splitting between the cells, the ratio of the measured extinction coefficients at 658 and 405 nm agreed with the literature cross section ratio to within 2.0 %. This excellent agreement provides strong evidence that there were no issues with contamination by absorbing gaseous or aerosol species during ozone calibrations.

Analysis of calibration data involved the following steps. Firstly, corrections were applied to normalise the raw microphone IA, $IA_{raw}$, by the laser power and cell resonance properties so that the calibration could later be applied to measured data with different laser powers and resonance properties. The corrected photoacoustic response, $IA_{norm}$, was calculated by multiplying $IA_{raw}$ by the correction factor shown in Eq. 3 (Arnott et al., 1999):

$$C = \frac{F_R}{QP_L}.$$ (3)

where $F_R$ is the cell resonance frequency, $Q$ the cell quality factor and $P_L$ the circulating laser power. $P_L$ was measured by the photodiode. PAS cell quality factors were in the range 87–93.

Secondly the background signal measured in the absence of ozone or particles, $IA_{norm}^{bkg}$, was characterised. This signal was subsequently subtracted from $IA_{norm}$ to yield the background corrected microphone signal, $IA_{corr}^{O3}$. A least-squares linear fit of $IA_{corr}^{O3}$ against CRDS-derived extinction was then performed to determine the PAS calibration coefficient. Figure 3 shows an example fit for a 405 nm PAS channel. Across all PAS cells, straight line gradients were typically in the range 0.02–0.32 and $R^2$ values were consistently >0.999. All regressions relating to the calibrations were forced through zero. A summary of the calibration gradients and $R^2$ values can be found in the supplementary material. The mean 1σ fitting uncertainty in the gradient of the linear ozone calibration gradients covering all cells was 0.15 %.

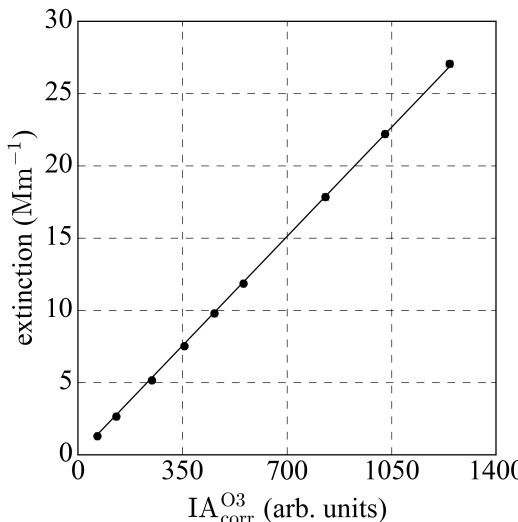

**Figure 3: Photoacoustic spectrometer response (microphone integrated area (IA)) and concurrent cavity ring-down spectrometer extinction coefficient for nine ozone concentrations at wavelength 405 nm. Each point is the mean of 90 seconds of 1Hz data and error bars represent the 2σ precision of each measurement; these are not discernible by eye due to the excellent precision.**

Since there was no green-wavelength CRDS cell available, the 514 nm PAS cell was calibrated by evaluating the 514 nm absorption coefficient using measurements from the 658 nm CRDS cell and Eq. 4:

$$\alpha_{abs\_514} = \frac{\alpha_{ext\_658}}{\sigma_{abs\_658}} \sigma_{abs\_514} \tag{4}$$

where $\sigma_{abs\_658} = 2.19\times10^{-21}$ cm$^2$ and $\sigma_{abs\_514} = 1.62\times10^{-21}$ cm$^2$ are the ozone absorption cross sections at the wavelengths of interest (Gorshelev et al., 2014), $\alpha_{ext\_658}$ is the extinction coefficient measured using the 658 nm CRDS

channel and $\alpha_{514}$ is the absorption measured by the 514 nm PAS channel. The 658 nm CRDS was used to calibrate the 514 nm PAS channel because it extended over a greater range of extinction coefficients (167–1506 Mm$^{-1}$) than the 405 nm CRDS (1–27 Mm$^{-1}$). This ensured that the 514 nm PAS calibration covered a range of absorption coefficients greater than that spanned by the nigrosin absorption coefficients. Calibrating the 405 nm channel using the 405 nm CRDS channel, as opposed to the 658 nm channel, would lead to absorption coefficients that were lower by 3.2 %. In the calculation of the

extinction coefficient (see Eq. 1), the Rayleigh scattering term is common to both the $\tau$ and $\tau_0$ measurements and therefore does not contribute to the extinction. Thus it is valid to scale the extinction coefficient measured with the CRDS at 658 nm (or 405 nm) by the literature absorption cross section ratio. What this analysis does not account for is any small difference in the Rayleigh scattering of air versus the Rayleigh scattering of air with a small ozone concentration (up to 680 ppm).

**2.4 Aerosol generation and conditioning**

Figure 4 shows a schematic diagram of the particle generation setup. Water-soluble nigrosin, a strong light-absorbing dye at visible wavelengths, (Sigma Aldrich, CAS Number 8005-03-6, lot number MKBR1705V, product number 198285-100G)

was dissolved into high purity deionised water (VWR Chemicals) with a range of concentrations between 3.2–7.1 grams per litre (g L$^{-1}$). The solution was drawn into a TSI constant output atomizer (TSI, 3076), which used high purity synthetic air (BOC, 270028-L) at a flow rate of approximately 2.5 L min$^{-1}$. The generated aerosol was dried to <10% relative humidity using a silica gel diffusion drier (Topas, DDU-570) and passed through an electrical ionizer (MSP, 1090). After exiting the ionizer, the aerosol stream was split between a differential mobility analyser (DMA) column (TSI, 3081) and mass flow controller. Flow rates through the mass flow controller were set to regulate the flow through the DMA such that the sample-to-sheath flow ratio was at least 1:10 with a sample flow rate in the range 0.3-0.4 L min$^{-1}$ and sheath flow rate in the range 3.5-4.0 L min$^{-1}$. This ensured that the flow through the ionizer was sufficiently high for a Fuch's charging distribution to be applied to the particles, while ensuring that the DMA could output particle diameters between 10 and 532 nm. The flow rate through the DMA decreased as its impactor removed particles with diameter >1 μm, which impeded the flow and thus altered the flow splitting between the mass flow controller (MFC) and DMA column. This varied by <5 % over the course of a test. Section 2.5 provides details of the sensitivity of modelled optical properties of nigrosin to DMA flow rates. The DMA was coupled to a CPC (TSI, 3776) to operate as a scanning mobility particle sizer (SMPS, path A in Fig. 4). This was used to characterise the atomizer output for periods at the start and end of each experiment, thus enabling quantification of any drift. To obtain a quasi-monodisperse aerosol size distribution for optical measurements, the DMA was operated at fixed voltages (path B in Fig. 4).

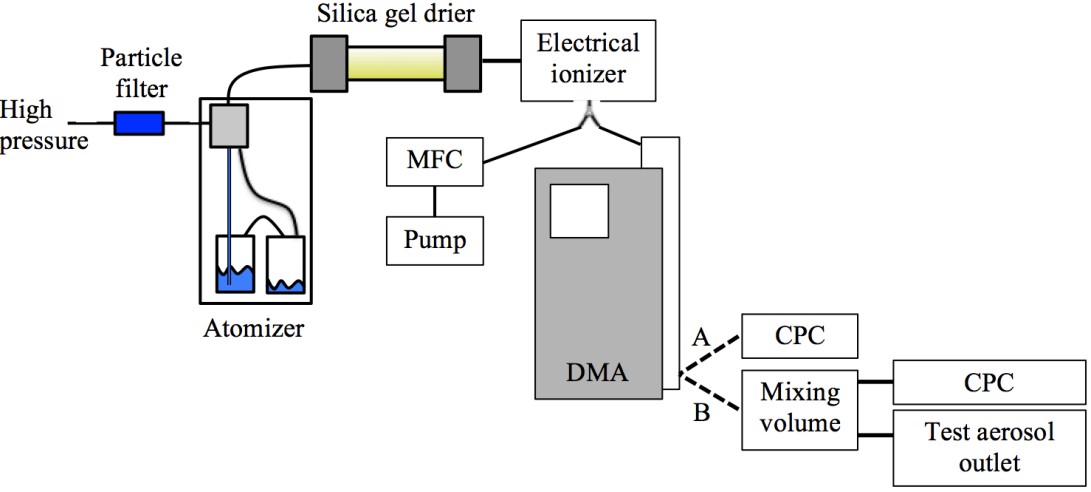

**Figure 4: Experimental setup for generating nigrosin aerosol. The dashed lines labelled 'A' and 'B' represent two independent flow lines (changed manually) used for scanning mode (line A) or fixed voltage size-selection mode (line B). Abbreviations: 'MFC': mass flow controllers, 'DMA': differential mobility analyser and 'CPC': condensation particle counter.**

For optical measurements, aerosolised nigrosin was mobility-selected with central diameters in the range 250–425 nm in 25 nm steps. The aerosol flow was split between optical cells using a series of Y-flow splitters (Brechtel, 1110 and 1104). All PAS cells were operated in parallel, with two of these cells also having CRDS channels in series as shown previously in Fig. 1. The flow rate through each cell was 1 L min$^{-1}$ and was controlled using mass flow controllers (Alicat, MC Series). This

resulted in a plug-flow residence time within each PAS cell of approximately 12 seconds. To measure the particle number concentration, the mass flow controller on the output of the 514 nm PAS cell was replaced with a second CPC (TSI, 3010), which used a critical orifice to control the flow rate to 1 L min$^{-1}$. Aerosol number concentrations during experiments ranged from 40–575 cm$^{-3}$.

The aerosol flow splitting between cells was evaluated to test whether the CPC-measured number count in the 514 nm PAS cell was representative of the other PAS cells. The aerosol number concentration within the mixing volume and at the outlet of each PAS cell was in turn measured using the two CPCs simultaneously. This allowed the particle transmission efficiency through each PAS cell to be determined independently of variations in particle generation stability. Differences between

particle transmission through the 514 nm PAS cell and other PAS cells ranged from 1.1–4.5 %. Particle concentrations were adjusted to account for these variations.

### 2.5 Modelling ensemble absorption cross sections

It is well established that the use of differential mobility analysis for aerosol size selection leads to the generation of a size distribution with polydispersity arising, in part, from the multiple charging of particles (e.g. Wiedensohler et al., 2012). To

correctly model the optical properties of the size-selected sample, the multiplet contributions must be taken into account. Transfer function theory predicts the aerosol size distribution exiting the DMA (Knutson and Whitby, 1975). Mie theory (Bohren and Huffman, 1998) can then be applied to calculate single particle optical cross sections at each diameter in the ensemble. By combining the two theories, the ensemble optical cross sections (hereafter referred to as the *cross section*) for quasi-monodisperse mobility-size-selected aerosol can be calculated. The following section describes how this was

implemented.

Firstly, we measured the polydisperse input aerosol size distribution to the DMA (Fig. 5(a)) using the SMPS. The accuracy of SMPS sizing was confirmed using monodisperse polystyrene latex spheres (ThermoFisher Scientific, 3000 Series Nanosphere Size Standard). Bimodal lognormal distributions were fitted to the SMPS-measured particle size distributions

over the diameter range 71–532 nm. A summary of the best-fit parameters can be found in the supplementary material. We measured size distributions before and after running mobility-selected nigrosin through the PAS and CRDS cells to characterise the variability in the particle size distribution. The impact that this variability had on the size distribution exiting the DMA and, consequently, on the modelled optical cross sections was evaluated by propagating each measured size distribution through the Mie closure routine (described below). In summary, variability in the aerosol source stability led to a

mean standard deviation in the modelled absorption cross sections of 4.3 % for all wavelengths and size-selected diameters.

The DMA transfer function describes the probability of a particle of given diameter exiting the DMA. It is used to derive the quasi-monodisperse size distribution at the DMA outlet when operated at fixed voltage (Knutson and Whitby, 1975). Figure

5(b) describes the diffusional transfer function for a particle with 250 nm mobility diameter calculated using the equations presented by Stolzenburg (1988). The transfer function was evaluated for the DMA geometry and aerosol/sheath flow rates used in this study. We verified our calculations of the transfer function in several ways. The width of the transfer function in the absence of diffusional broadening was verified using the expression presented by Stolzenburg and McMurry (2008). The diffusional broadening parameters used in the transfer function model were also evaluated against the values presented in Stolzenburg (1988), namely the $G_{DMA}$ and $\widetilde{D}$ factors, which agreed to better than 1 %. Finally, the diffusional transfer function was verified quantitatively against the Hagwood et al. (1999) simulations, which also used the same Stolzenburg (1988) formulation as used in this work.

To model the quasi-monodisperse aerosol size distribution at the DMA outlet when operated at a fixed voltage, the particle charging efficiencies for the range of particle diameters in the polydisperse input size distribution were calculated using Fuch's charging theory (Wiedensohler, 1988). Aerosol charging efficiencies were calculated for positive ions with up to six elementary charges (Fig. 5(c)). This was found to be a good approximation of particle charging efficiencies for the experimental setup described in Fig. 1. We verified that the modelled ratios of singly to doubly charged particles exiting the DMA agreed with experimentally measured ratios using polydisperse size distributions from a Passive Cavity Aerosol Spectrometer Probe (DMT, PCASP-100X). Modelled and measured charge fractions agreed to within 6 % for particles with diameter $\geq$ 250 nm, with this uncertainty in part due to the resolution of the PCASP diameter bins. Propagating this uncertainty through the Mie closure routine (described below) led to a mean uncertainty of ±0.93 % in the modelled absorption cross sections for all wavelengths and mobility-selected diameters.

The quasi-monodisperse aerosol size distribution, i.e. the size distribution exiting the DMA, was calculated by multiplying the polydisperse aerosol size distribution at the DMA input by the DMA diffusional transfer function and the aerosol charging efficiencies for corresponding particle diameters (Fig. 5(d)).

From the calculated size distribution exiting the DMA, the absorption and extinction cross sections were calculated using Mie theory (Bohren and Huffman, 1998) for each particle diameter in the size distribution, the PAS wavelengths and nigrosin refractive indices from Bluvshtein et al. (2017) (Fig. 5(e)). Mie theory assumes that a particle interacting with radiation is spherical, which is a reasonable assumption for nigrosin particles based on previous studies (e.g. Lack et al., 2006). We chose to use the refractive index values reported by Bluvshtein et al. (2017) to facilitate direct comparison between the two sets of results. The refractive indices used in this analysis were 1.624±0.0063 + (0.1541±0.0081)$i$ for 405 nm, 1.622±0.0085 + (0.2594±0.011)$i$ for 514 nm and 1.811±0.007 + (0.2476±0.0031)$i$ for 658 nm. Sensitivity of the modelled absorption cross section to the imaginary part of the refractive index was quantified using the values and uncertainties presented in Bluvshtein et al. (2017), resulting in a mean uncertainty for all wavelengths and mobility-selected

diameters of 1.15 %. Similarly, uncertainty in the modelled absorption cross sections due to a ±5 % change in the DMA aerosol flow rate was 0.21 %.

Finally, the ensemble absorption and extinction cross sections were calculated by weighting the Mie cross sections by the
relative number of each size particle exiting the DMA, calculated using

$$\overline{\sigma_{abs}} = \sum_i \sigma_i^{abs} N_i ,\qquad(5)$$

where $\sigma_i^{abs}$ is the Mie absorption cross section at diameter $D_i$ and $N_i$ is the component of the normalised size distribution at diameter $D_i$, i.e. the distribution that was assumed to enter the PAS and CRDS cells, such that $\sum N_i = 1$. A similar expression was used to calculate $\overline{\sigma_{ext}}$ where $\sigma_{abs}$ was replaced by $\sigma_{ext}$. The cumulative absorption cross section for 250 nm
mobility-selected nigrosin is plotted in Fig. 5(f). This highlights the relative importance of the absorption contribution from multiply charged particles. Although this contribution was lower for larger mobility-selected diameter particles, it can still significantly contribute to absorption as shown by the dashed green line in Fig. 5(f) for 400 nm diameter particles.

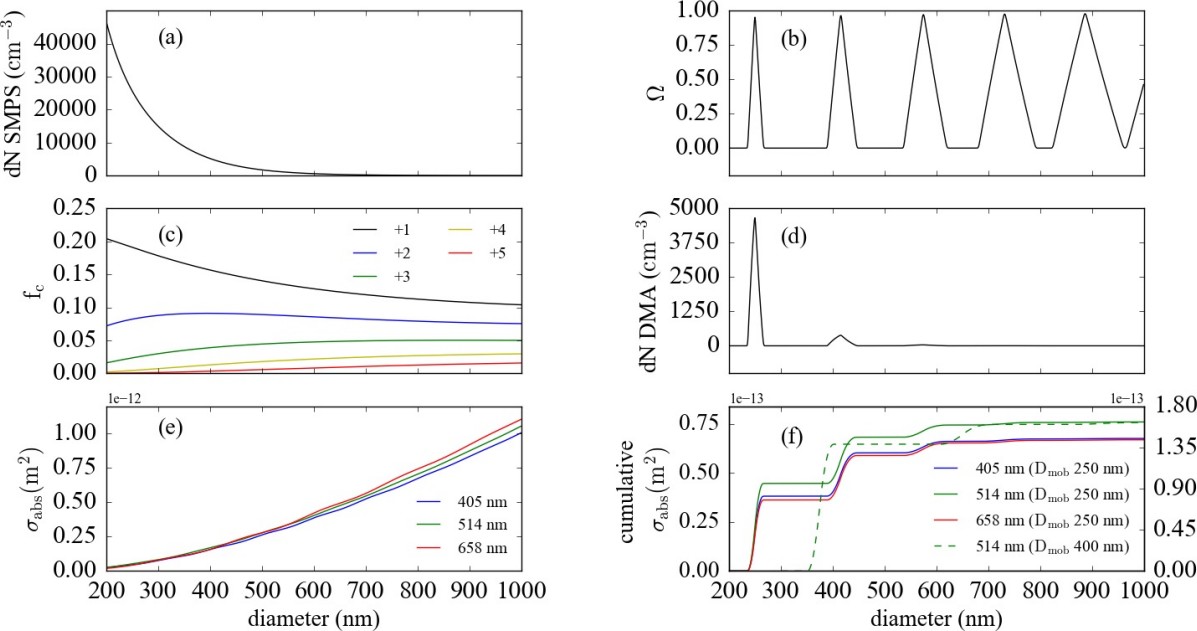

**Figure 5: An overview of steps involved in modelling the absorption cross section. (a) SMPS-measured particle size distribution. (b) DMA diffusional transfer function for a fixed DMA voltage corresponding to a mobility-selected diameter of 250 nm. (c) Fuch's charging probabilities for positively charged particles. The figure legend indicates the magnitude of the positive charge. (d) Modelled size distribution exiting the DMA column when operated at fixed voltage for a mobility diameter of 250 nm. (e) Absorption cross sections calculated using Mie theory for three visible wavelengths of light, as indicated in the figure legend. (f)**
**Cumulative absorption cross sections for nigrosin with a mobility-selected diameter of 250 nm. The dashed green line represents**

the absorption cross section at wavelength 514 nm corresponding to a 400 nm mobility-selected diameter nigrosin particle using the scale on the right.

## 3 Results and discussion

Initially, we verified the accuracies of the cavity ring-down spectrometers, as they form an integral part of the photoacoustic spectrometer calibrations. The ensemble extinction cross sections (hereafter referred to as extinction cross section) for nigrosin with mobility-selected diameters in the range 250–425 nm were measured using CRDS and modelled using Mie theory, as outlined in Sect. 2.5. The mean gradient between the modelled and CRDS-measured extinction cross sections was $0.98 \pm 0.01$ ($2\sigma$ fitting uncertainty) as shown in Fig. 6. Gradients for the 658 nm and 405 nm wavelengths were 0.96 and 1.00 respectively.

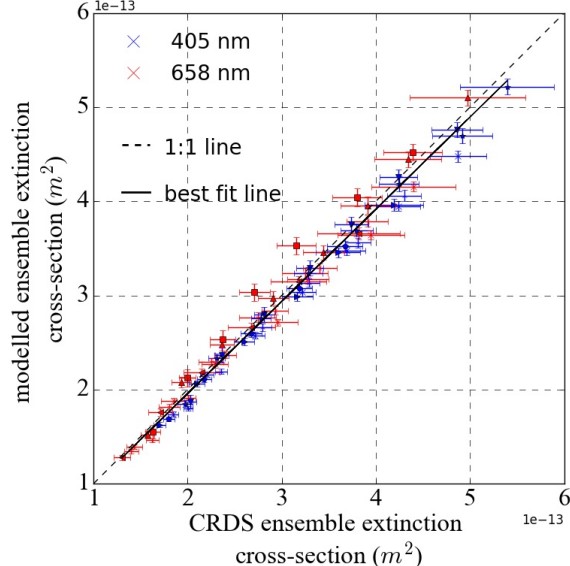

**Figure 6: Modelled versus CRDS-measured extinction cross sections for nigrosin aerosol at 405 nm and 658 nm wavelengths.**

The mean gradient between the modelled and PAS-measured absorption cross sections for nigrosin for all five ozone-calibrated PAS cells was $1.08 \pm 0.01$ ($2\sigma$ fitting uncertainty) as shown in Fig 7. Gradients for the 405, 514 and 658 nm wavelengths were 1.08, 1.07 and 1.09 respectively. These data encompass multiple experimental runs using three independent ozone calibrations.

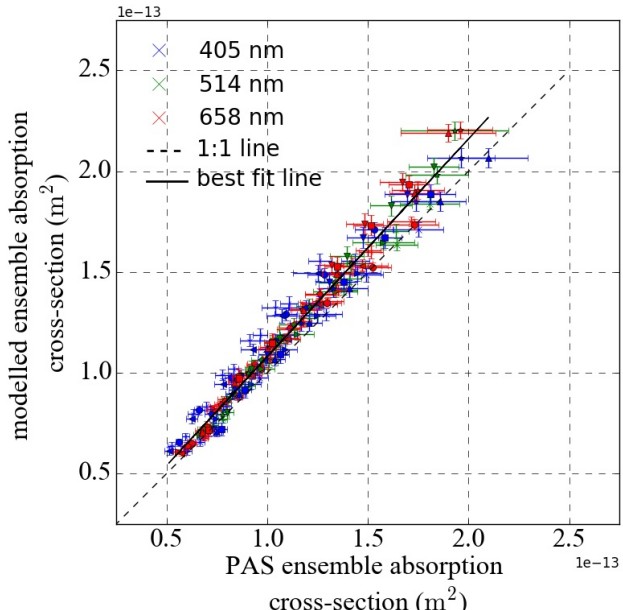

**Figure 7: Modelled versus PAS-measured absorption cross sections for nigrosin aerosol at 405, 514 and 658 nm wavelengths.**

The uncertainties in the measured optical cross sections for nigrosin shown by the error bars in Fig. 6 and Fig. 7 represent the standard deviations for each 90-second cross section measurement, encompassing the precision in both the associated PAS absorption coefficient and CPC measurements. Modelled optical cross sections may be subject to an additional 5 % bias due to uncertainty in the CPC accuracy (Fletcher et al., 2009). Similarly, uncertainties in the modelled cross sections were calculated by combining the uncertainties due to variability in the size distribution, charging distribution and imaginary part of the refractive index in quadrature.

The key result of this work is the demonstration that photoacoustic spectrometers can be accurately calibrated using ozone at short wavelengths (405 nm), which contrasts with the recent results of Bluvshtein et al. (2017). A thorough evaluation of model uncertainties has shown that this result is robust, despite uncertainties in the imaginary part of the nigrosin refractive index and variability in the measured polydisperse aerosol size distribution used to calculate model absorption properties. It is unclear as to the underlying cause of discrepancy between results presented here and those in Bluvshtein et al. (2017). Possible reasons could include measurement contamination or differences in methods used for the calculation of model optical cross sections. In particular, we have demonstrated the importance of accurately modelling the contribution of multiply charged particles to capture the optical behaviour of the quasi-monodisperse distributions used in these experiments.

**4 Conclusions**

This study has shown that the ozone calibration method can be used to calibrate accurately photoacoustic instruments operating at short-visible wavelengths. This alleviates concerns based on previously published results, which have shown large discrepancy at 404 nm. Using nigrosin aerosol with mobility-selected diameters in the range 250–425 nm, we verified

that the measured absorption cross sections using photoacoustic spectroscopy agreed with modelled values to within 8 %. Our result is robust for optical wavelengths 405, 514 and 658 nm.

*Data availability.* For data related to this paper please contact Justin Langridge (justin.langridge@metoffice.gov.uk).

*Competing interests.* The authors declare that they have no conflict of interest.

*Acknowledgements.* This work was funded by the Met Office. In addition, NWD was supported by a NERC/Met Office Industrial Case studentship (Ref 640052003). MIC was supported by an Analytical Chemistry Trust Fund Tom West Fellowship. MIC and JMH were supported by the CLARIFY-2017 Natural Environment Research Council funded proposal (NE/L013797/1). We thank Professor Andrew Orr-Ewing and Philip Coulter of the University of Bristol for their help in measurements of the spectrum of the laser sources used in this work.

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
