# Peer review of "On the accuracy of aerosol photoacoustic spectrometer calibrations using absorption by ozone"

_Atmospheric Measurement Techniques, 2017_

## Referee Comment (RC1) · Anonymous Referee #1 · 6 Feb 2018

Davies et al. (2018) present work on calibrating photoacoustic spectrometers using ozone, and compare aerosol optical properties determined using the ozone calibrations to optical properties measured using a different method. The author find that the ozone calibration works well at 405 nm, and that the measured absorption coefficients for nigrosin aerosol agree well with absorption coefficients calculated using refractive indices derived from ellipsometry. This is in contrast to previous work that found that using ozone to calibrate the PAS resulted in absorption coefficients twice as large as those derived from the ellipsometry measurements. Additionally, unlike the previous work, the authors performed experiments at three wavelengths, and found excellent agreement at all three wavelengths. This is a valuable study, and I recommend that it

be published in AMT.

My main concern is that no effort was made to verify that ozone was the only source of extinction/absorption in the CRDS and PAS cells. While the agreement at all three wavelengths between the measured aerosol absorption and the absorption calculated using the previously measured refractive indices argues against an interference, it would be nice to have some additional checks, such as measuring the particle number concentrations during the ozone calibration, measuring the ozone by an independent method and comparing the extinction calculated using that concentration to the measure extinction, or putting the same ozone flow into the blue and red CRDS cells and making sure the concentrations calculated using the literature cross sections agree. This last check will be somewhat tricky due to the large difference in the ozone cross sections, but it should be possible in the 30-50 ppmv range, and it would provide a nice check to ensure that there are no other sources of extinction such as $NO_2$ (or possibly another absorbing gas) or aerosol particles in the cells. For the 405 channel, reversing the order of the CRDS and PAS cells could also be tried.

I recognize that the authors may not have all the equipment needed to perform some of these experiments (e.g. an ozone monitor), and so not all of these experiments need to be performed prior to publication. However, the authors should be able to do some of these checks using the data they already have (such as comparing calculated ozone concentrations from the blue and red CRDS), and given the discrepancy between this study and the study by Bluvshtein et al. (2017), it would be nice to see these additional checks performed to help investigators going forward.

**Specific comments:**

1. P2L29-P3L5: Somewhere in the introduction or possibly in the discussion section it would be good to mention the recent work by Cremer et al. (2017), who found

that the photoacoustic response was lower than would be expected based on Mie calculations, and how those results relate to yours.

2. P4L9-10: "The cell was positioned within a multi-pass optical system formed by two cylindrical mirrors. . . " Approximately how many passes does the laser make? Also, it should be mentioned that the concavities of the two mirrors are rotated 90° to each other.

3. P4L12: How did you measure the laser wavelengths and line widths?

4. P4L15: What is the manufacturer and part number for the microphones?

5. P4L23-25: "Aerosol absorption coefficients ($m^{-1}$) measured by the photoacoustic spectrometers were converted to absorption cross sections ($m^2$) for comparison to theoretical calculations by dividing by the aerosol number concentrations reported by a CPC (see Sect. 2.4)." I interpret this to mean you divided the measured absorption by the measured concentration, without correcting for the presence of multiply charged particles. If so, these cross sections should be referred to as **effective** cross sections, since the cross sections you would get from this method are going to be larger than what you would calculate from Mie theory due to the presence of multiply charged particles. There are several other places where this applies.

6. P4L26-27: "Cavity ring-down spectroscopy is a highly sensitive technique used for measuring the optical extinction coefficient of gases and particulate matter (O'Keefe and Deacon, 1988; Romanini et al., 1997) without the need for instrument calibration." I'm not sure it's 100% correct to say that CRDS does not require calibration (e.g. Toole et al., 2013). The raw CRDS signal also needs to be adjusted to take into account $R_L$, which can require calibration (see below).

7. P4L32: Please provide ring-down time constants for the two CRDS channels.

8. P5L3-4: What is the radius of curvature of the CRDS mirrors?

9. P5L12-13: "Cavity mirror-to-mirror lengths ranged from 371-423 mm yielding $R_L$ factors in the range 1.150-1.173." How were the $R_L$ values measured? While determining $R_L$ using the physical dimensions of the CRDS cell may be appropriate for aerosol particles (Langridge et al., 2011), Fuchs et al. (2008) found that for gases $R_L$ was not equal to the geometric $R_L$. Were any experiments performed to determine if $R_L$ in your system is different for gas and particles?

   Is there a reason for the different cavity lengths?

10. P5L14-15: See comment 5.

11. P5L20: Were the CRDS cells made of teflon or metal? If they were metal, please specify the material.

12. P6 Figure 1: This figure was hard to understand at first because I expected the colors of a given box to correspond to the wavelength of that instrument. I think the figure would be clearer if the colors of the PAS and CRDS cells corresponded to the wavelength used for that cell. Perhaps then use different shapes to differentiate between the CRDS and PAS cells?

13. P6L7: "...the measured ozone concentrations were used directly." How were the ozone concentrations measured (see comment below), or do you mean measured extinctions were used directly?

14. P6L13-14: "At the start of each calibration cycle, pure oxygen was introduced into the PAS cells through the ozone manifold. The oxygen displaced a fraction of the filtered-air flow through each cell..." This sentence and Eq. 2 imply that the flow through the system was a mix of air and gas from the ozone generator. Is this correct? If so, what fraction of the flow came from the ozone generator? How was the ambient air filtered?

15. P6L20: "The 515 nm PAS cell..." Isn't the wavelength 514 nm?

16. P6L21-22: "Ozone splitting ratios derived using this method compared extremely well to in-line mass flow measurements and were in the range 2-28%." What do you mean by "2-28%." Do you mean that the difference in the flow between two cells was between 2 and 28%, or that the splitting ratio, $\Delta\nu$, calculated from Eq. 2 was between 2 and 28%?

17. P7L7-9 and P8 Figure 3: "Ozone concentrations in the range $\sim$10-500 ppm were used." How did you determine the ozone concentrations? Figure 3 shows a maximum extinction of 27 Mm$^{-1}$ at 405 nm. If this is only from ozone, this gives an ozone concentration of $\sim$660-750 ppmv ($\sigma_{ozone}$ around 405 nm is $1.45 - 1.65 \times 10^{-23}$ cm$^2$, depending on the exact wavelength (Serdyuchenko et al., 2014)), higher than the 500 ppmv in the text. Also, 10 ppmv of ozone gives an extinction of $\sim$50 Mm$^{-1}$ at 658 nm. Did you put lower ozone concentrations into the green and red PAS cells to extend the calibration curves to lower values?

    Were the same ozone levels used for both the 405 and 658 CRDS channels? If so, how do the ozone concentrations calculated using the measured extinction and the literature cross sections compare for those two wavelengths?

18. P7L17 (Eq. 3): Why is the resonant frequency represented by $\nu$ in this equation, and $F_r$ in Eq. 2?

19. P7L18: How is $P_L$ measured? By the photodiode? Also, what are typical quality factors for your instrument?

20. P8L1 (and elsewhere): I might consider replacing "gradient" with the more common "slope," but this is mostly preference on my part.

21. P8 Figure 3: The y-axis units are inverse megameters (Mm$^{-1}$), while the units in the text (e.g. P4L23 and P5L9) are inverse meters (m$^{-1}$). Since Mm$^{-1}$ are the

customary units in aerosol work, I would suggest changing the units in the text, but whatever you choose, the units should be consistent.

22. P8L10: Why was the red CRDS used to calibrate the green PAS? Do you get the same result if you use the blue CRDS instead?

23. P8L17: Please provide the product number and lot number for the nigrosin used.

24. P9L2: What was the DMA sheath flow?

25. P9L11: "The aerosol flow was split between optical cells using a series of Y-flow splitters." Please show how this was done in Figure 1.

26. P12 Figure 5e-f: When I use either the Bohren and Huffman Mie codes or an online Mie calculator (https://omlc.org/calc/mie_calc.html) to calculate absorption cross sections, I get numbers lower than those shown in Figure 5e. For example, for 1000 nm diameter particles, I get the following absorption cross sections: $1.01 \times 10^{-12}$ m$^2$ at 405 nm; $1.06 \times 10^{-12}$ m$^2$ at 514 nm; and $1.10 \times 10^{-12}$ m$^2$ at 658 nm. The absorption cross section that I calculate for 400 nm particles at a wavelength of 514 nm ($1.69 \times 10^{-13}$ m$^2$) is higher than the value shown in Figure 5f ($\sim 1.4 \times 10^{-13}$ m$^2$ ) Please explain these discrepancies. If there was an error in the Mie calculation, how does this affect the agreement between the measured and modeled aerosol extinction and absorption cross sections?

27. P12L19-20: See comment 5.

28. P13 Figure 6 and P14 Figure 7: Are the measured cross sections effective cross sections (measured extinction or absorption divided by the particle concentration) or are they corrected for the effect of multiply charged particles?

29. Supplementary material, Tables S1 and S2: What do "gdry," "rtd," "btd," and the other abbreviations in the second column of Table S2 stand for? I'm assuming

that these are the names of the different CRD/PAS cells. If so, the names in the text, in Figure 1, and in the supplement should all be consistent.

30. Supplementary material, Table S2: I'm assuming that "bdry" and "btd" refer to the two 405 PAS cells. If so, why is the gradient for the bdry/bdry PAS/CRDS calibration 25% higher than the gradient for the btd/bdry PAS/CRDS calibration (and the same for rtd/rdry and rdry/rdry)? If you put the blue CRDS in front of PAS 1 instead of PAS 5, do you get the same gradients?

**Technical corrections:**

- P2L34, P3L30, P4L3, P5L14, P11L28 (and possibly other places): There's an extra comma after the "et al." in the in-text citation, e.g. Bluvshtein et al., (2017) should be Bluvshtein et al. (2017).

- P2L29 and P17L33: The second "m" in McManus should be capitalized.

- P16L31-32, P17L33-34: Please provide the DOI for these references (and any other references where the DOI was omitted).

**References**

Bluvshtein, N., Flores, J. M., He, Q., Segre, E., Segev, L., Hong, N., Donohue, A., Hilfiker, J. N., and Rudich, Y.: Calibration of a multi-pass photoacoustic spectrometer cell using light-absorbing aerosols, Atmospheric Measurement Techniques, 10, 1203-1213, https://doi.org/10.5194/amt-10-1203-2017, 2017.
Cremer, J. W., Covert, P. A., Parmentier, E. A., and Signorell, R.: Direct Measurement of Photoacoustic Signal Sensitivity to Aerosol Particle Size, The Journal of Physical Chemistry Letters, 8, 3398-3403, https://doi.org/10.1021/acs.jpclett.7b01288, 2017.

Davies, N. W., Cotterell, M. I., Fox, C., Szpek, K., Haywood, J. M., and Langridge, J. M.: On the accuracy of aerosol photoacoustic spectrometer calibrations using absorption by ozone, Atmospheric Measurement Techniques Discussions, 2018, 1-19, https://doi.org/ 10.5194/amt-2017-434, 2018.

Fuchs, H., Dube, W. P., Ciciora, S. J., and Brown, S. S.: Determination of Inlet Transmission and Conversion Efficiencies for in Situ Measurements of the Nocturnal Nitrogen Oxides, $NO_3$, $N_2O_5$ and $NO_2$, via Pulsed Cavity Ring-Down Spectroscopy, Analytical Chemistry, 80, 6010-6017, https://doi.org/10.1021/ac8007253, 2008.

Langridge, J. M., Richardson, M. S., Lack, D., Law, D., and Murphy, D. M.: Aircraft Instrument for Comprehensive Characterization of Aerosol Optical Properties, Part I: Wavelength-Dependent Optical Extinction and Its Relative Humidity Dependence Measured Using Cavity Ringdown Spectroscopy, Aerosol Science and Technology, 45, 1305-1318, https://doi.org/10.1080/02786826.2011.592745, 2011.

Serdyuchenko, A., Gorshelev, V., Weber, M., Chehade, W., and Burrows, J. P.: High spectral resolution ozone absorption cross-sections-Part 2: Temperature dependence, Atmospheric Measurement Techniques, 7, 625-636, https://doi.org/10.5194/amt-7-625-2014, 2014.

Toole, J. R., Renbaum-Wolff, L., and Smith, G. D.: A Calibration Technique for Improving Refractive Index Retrieval from Aerosol Cavity Ring-down Spectroscopy, Aerosol Science and Technology, 47, 955-965, https://doi.org/10.1080/02786826.2013.805875, 2013.

---

## Referee Comment (RC2) · Anonymous Referee #2 · 14 Feb 2018

This is a very interesting and important manuscript that seeks to explore the problems with ozone calibration experienced by Bluvshtein et al. (2017). It is suitable for publication in AMT and should be published after the following comments are taken into account:

1. P1L10-11: "Photoacoustic instruments require calibration, which is often achieved by measuring the photoacoustic signal generated by known quantities of gaseous ozone." I'm not sure how often ozone is really used to calibrated photoacoustic instruments. Please quantify or replace with "Photoacoustic instruments require calibration, which can be achieved by measuring the photoacoustic signal generated by known

quantities of gaseous ozone."

2. The ozone calibration of photoacoustic instruments for the measurement of aerosol absorption coefficients needs to put into the context of the calibration of such instruments with aerosols and other calibration gases. I recommend adding a short paragraph to the introduction. The following references, in addition to those already in the manuscript, come to mind: (Arnott et al., 2000;Gillis et al., 2010;Nakayama et al., 2015;Tian et al., 2009).

3. P4L28: "without the need for instrument calibration". This is not entirely correct; one needs to calibrate for mirror losses and the effective cavity length needs to be determined especially as the mirrors are purged with clean air.

4. P5L3-4: Please give the radius of curvature of the cavity mirrors.

5. P5L13: Please explain how the RL factors were determined.

6. P5L14-15: Replace "extinction cross sections" with "average extinction cross sections".

7. P5L19-20: "Teflon tubing was used throughout the flow system to minimize contamination.". Add "and reduce ozone losses". Please also specify the material used for the insides of the CRDS and PAS cells.

8. P8L1-3: Please also discuss the zero-offset of the linear regressions here and elsewhere unless the regressions were forced through zero; if this is the case please note this.

9. P8L11 Eq. 4: This seems to assume that the wavelength dependence of absorption equals that of extinction. How large is the influence of scattering (Rayleigh plus particle contamination)?

10. P15L6: "Our result is robust for optical wavelengths between 405 and 658 nm." This seems to be overstating the results as measurements at only one wavelength

(i.e., 514 nm) between 405 and 658 nm were discussed.

REFERENCES

Arnott, W. P., Moosmuller, H., and Walker, J. W.: Nitrogen Dioxide and Kerosene-Flame Soot Calibration of Photoacoustic Instruments for Measurement of Light Absorption by Aerosols, Rev. Sci. Instrum., 71, 4545-4552, 2000.

Bluvshtein, N., Flores, J. M., He, Q., Segre, E., Segev, L., Hong, N., Donohue, A., Hilfiker, J. N., and Rudich, Y.: Calibration of a Multi-Pass Photoacoustic Spectrometer Cell Using Light-Absorbing Aerosols, Atmos. Meas. Tech., 10, 1203-1213, 10.5194/amt-10-1203-2017, 2017.

Gillis, K. A., Havey, D. K., and Hodges, J. T.: Standard Photoacoustic Spectrometer: Model and Validation Using O2 A-band spectra, Rev. Sci. Instrum., 81, 064902, 2010.

Nakayama, T., Suzuki, H., Kagamitani, S., Ikeda, Y., Uchiyama, A., and Matsumi, Y.: Characterization of a Three Wavelength Photoacoustic Soot Spectrometer (PASS-3) and a Photoacoustic Extinctiometer (PAX), J. Meteorol. Soc. Jpn., 93, 285-308, 10.2151/jmsj.2015-016, 2015.

Tian, G., Moosmuller, H., and Arnott, W. P.: Simultaneous Photoacoustic Spectroscopy of Aerosol and Oxygen A-band Absorption for the Calibration of Aerosol Light Absorption Measurements, Aerosol Sci. Tech., 43, 1084-1090, 2009.

---

## Author Comment (AC2) · 27 Mar 2018

**On the accuracy of aerosol photoacoustic spectrometer calibrations using absorption by ozone**

Nicholas W. Davies, Michael I. Cotterell, Cathryn Fox, Kate Szpek, Jim M. Haywood, and Justin M. Langridge

We would like to thank the reviewers for taking the time to read our manuscript thoroughly and for highlighting some important issues, which will be addressed in turn below.

**Review 2**

1. P1L10-11: "Photoacoustic instruments require calibration, which is often achieved by measuring the photoacoustic signal generated by known quantities of gaseous ozone." I'm not sure how often ozone is really used to calibrated photoacoustic instruments. Please quantify or replace with "Photoacoustic instruments require calibration, which can be achieved by measuring the photoacoustic signal generated by known quantities of gaseous ozone."

We have modified the following sentence in the manuscript (P1L10-11):

*"Photoacoustic instruments require calibration, which can be achieved by measuring the photoacoustic signal generated by known quantities of gaseous ozone."*

2. The ozone calibration of photoacoustic instruments for the measurement of aerosol absorption coefficients needs to put into the context of the calibration of such instruments with aerosols and other calibration gases. I recommend adding a short paragraph to the introduction. The following references, in addition to those already in the manuscript, come to mind: (Arnott et al., 2000;Gillis et al., 2010;Nakayama et al., 2015;Tian et al., 2009).

We have added the following to the manuscript (P2L33-P3L3):

*"There exist a number of options for calibrating photoacoustic spectrometers including use of nitrogen dioxide (Atnott et al., 2000; Nakayama et al., 2015), polydisperse kerosene soot (Nakayama et al., 2015), oxygen (Tian et al., 2005; Gillis et al., 2010) and ozone (Lack et al. 2006; Lack et al., 2012; Bluvshtein et al., 2017). Ozone was chosen as the calibrant for our PAS cells, in part as nitrogen dioxide has been shown to introduce uncertainty in calibrations at 405 nm due to photolysis (Lack et al., 2012) and generation of aerosol particles is challenging in the field."*

3. P4L28: "without the need for instrument calibration". This is not entirely correct; one needs to calibrate for mirror losses and the effective cavity length needs to be determined especially as the mirrors are purged with clean air.

We have modified the following sentence in the manuscript (P5L10-11) such that it now reads:

*"Cavity ring-down spectroscopy is a highly sensitive technique used for measuring the optical extinction coefficient of gases and particulate matter (O'Keefe and Deacon, 1988; Romanini et al., 1997)."*

We have also determined the CRDS $R_L$ factor for ozone (please refer to our response to reviewer 1, comment 9).

4. P5L3-4: Please give the radius of curvature of the cavity mirrors.

We have modified the following sentence in the manuscript (P5L17-19):

*"Cavity mirrors were manufactured from fused silica with wavelength-specific coatings, 25 mm diameter, 1m radii of curvature and reflectivities in excess of 99.99 % (Layertec GmbH, red 660 nm; CVI Laser Optics, blue 405 nm)."*

5. P5L13: Please explain how the RL factors were determined.

The following was added to the manuscript (P5L31-P6L7):

*"The $R_L$ factor appropriate for aerosol measurements was determined from the geometric dimensions of the detection cell. As highlighted by Fuchs et al. (2008), the $R_L$ factor for detection of gaseous species can be different from this value, due to ability of gaseous samples to diffuse. We determined the gaseous $R_L$ factors by measuring the change in the ring-down times for filtered air plus ozone in (i) standard operation whereby ozone partially diffuses into the volume between the sample inlet and mirror and (ii) non-standard operation whereby ozone was fully mixed into the volume between the sample inlet and mirror by pulling the ozone-laden air out of the cavity through the mirror purge lines. This resulted in $R_L$ factors 1.05 (658 nm) and 1.04 (405 nm)."*

These different $R_L$ factors have been propagated through the analysis (please refer to our response to reviewer 1, comment 9).

6. P5L14-15: Replace "extinction cross sections" with "average extinction cross sections".

We have reworded the following in the manuscript (P6L8-10) so that it now

reads:

*"Extinction coefficients were converted to ensemble extinction cross sections ($m^2$) by dividing by the aerosol number concentrations measured using a CPC (see Sect. 2.4). The ensemble extinction cross section is hereafter referred to as the extinction cross section."*

7. P5L19-20: "Teflon tubing was used throughout the flow system to minimize contamination.". Add "and reduce ozone losses". Please also specify the material used for the insides of the CRDS and PAS cells.

We have modified the following sentence in the manuscript (P6L14-15):

*"Teflon tubing was used throughout the flow system to minimise contamination and to reduce ozone losses."*

We have also added the following sentence to the manuscript (P5L17-18):

*"The CRDS cells were manufactured from aluminium."*

The PAS cells were manufactured out of Aluminium. Please refer to P4L12.

8. P8L1-3: Please also discuss the zero-offset of the linear regressions here and elsewhere unless the regressions were forced through zero; if this is the case please note this.

The following sentence has been added to the manuscript (P9L25):

*"All regressions relating to the calibrations were forced through zero."*

9. P8L11 Eq. 4: This seems to assume that the wavelength dependence of absorption equals that of extinction. How large is the influence of scattering (Rayleigh plus particle contamination)?

We have added the following to the manuscript (P10L14-18):

*"In the calculation of the extinction coefficient (see Eq. 1), the Rayleigh scattering term is common to both the $\tau$ and $\tau_0$ measurements and therefore does not contribute to the extinction. Thus it is valid to scale the extinction coefficient measured with the CRDS at 658 nm (or 405 nm) by the literature absorption cross section ratio. What this analysis does not account for is any small difference in the Rayleigh scattering of air versus the Rayleigh scattering of air with a small ozone concentration (up to 680 ppm)."*

10. P15L6: "Our result is robust for optical wavelengths between 405 and 658 nm." This seems to be overstating the results as measurements at

only one wavelength (i.e., 514 nm) between 405 and 658 nm were discussed.

We have modified the following sentence in the manuscript (P17L6):

*"Our result is robust for the optical wavelengths 405, 514 and 658 nm."*

---

## Author Comment (AC1)

**On the accuracy of aerosol photoacoustic spectrometer calibrations using absorption by ozone**

Nicholas W. Davies, Michael I. Cotterell, Cathryn Fox, Kate Szpek, Jim M. Haywood, and Justin M. Langridge

We would like to thank the reviewers for taking the time to read our manuscript thoroughly and for highlighting some important issues, which will be addressed in turn below.

**Review 1**

 P2L29-P3L5: Somewhere in the introduction or possibly in the discussion section it would be good to mention the recent work by Cremer et al. (2017), who found that the photoacoustic response was lower than would be expected based on Mie calculations, and how those results relate to yours.

We have added the following to the manuscript (P2L29-31):

"Biases associated with PAS include a lack of proportionality between the photoacoustic signal and the aerosol absorption cross section for particles with radii greater than 0.7  $\mu$ m (Cremer et al., 2017). This is not an issue for the current study, which uses an impactor to remove particles with radii > 0.5  $\mu$ m; see Sect. 2.4."

2. P4L9-10: "The cell was positioned within a multi-pass optical system formed by two cylindrical mirrors. . . " Approximately how many passes does the laser make? Also, it should be mentioned that the concavities of the two mirrors are rotated  $90^{\circ}$  to each other.

We have added the following to the manuscript (P4L16):

"The concavities of the two mirrors were rotated 90° to each other."

We have also added the following to the manuscript (P4L22-27):

"In an optimally aligned system, the laser would pass through the acoustic resonator 182 times (Silver et al., 2005; Lack et al., 2012). However, no effort was made to achieve this limit in the current system. Alignment was conducted by visual inspection of the spot pattern only, which almost certainly resulted in a lower number of passes. Quantifying the number of passes through the resonator was not critical. Light exiting the resonator was measured using a photodiode, which allowed the PAS signal to be corrected for any laser power or alignment instability (Lack et al., 2012)."

3. P4L12: How did you measure the laser wavelengths and line widths?

We have added the following to the manuscript (P4L19-20):

"Laser wavelengths and line widths were measured using an Avantes spectrometer (CompactLine) for the blue and green wavelengths and a Hamamatsu spectrometer (C11697MB) for red wavelengths."

4. P4L15: What is the manufacturer and part number for the microphones?

We have modified the following in the manuscript (P4L27-29):

"The acoustic signal was detected using microphones (Knowles Acoustics, EK-23132) positioned half way along the lengths of each resonator to coincide with the pressure antinode corresponding to the lowest-order (n = 1) acoustic eigenmode of the photoacoustic cell."

5. P4L23-25: "Aerosol absorption coefficients  $(m^{-1})$  measured by the photoacoustic spectrometers were converted to absorption cross sections

 $(m^2)$  for comparison to theoretical calculations by dividing by the aerosol number concentrations reported by a CPC (see Sect. 2.4)." I interpret this to mean you divided the measured absorption by the measured concentration, without correcting for the presence of multiply charged particles. If so, these cross sections should be referred to as **effective** cross sections, since the cross sections you would get from this method are going to be larger than what you would calculate from Mie theory due to the presence of multiply charged particles. There are several other places where this applies.

The reviewer has interpreted this correctly. Although we define 'ensemble cross sections' further into the manuscript, it is important to highlight this here too. Hence we have reworded this in the manuscript (P5L3-8) such that it now reads:

"Aerosol absorption coefficients (Mm-1) measured by the photoacoustic spectrometers were converted to ensemble absorption cross sections (m2) for comparison to theoretical calculations by dividing by the aerosol number concentrations reported by a CPC (see Sect. 2.4). The ensemble absorption cross section represents the mean of the absorption cross sections corresponding to a range of particles sizes, for example from multiply charged particles (see Sect. 2.5). The ensemble absorption cross section is hereafter referred to as the absorption cross section." 6. P4L26-27: "Cavity ring-down spectroscopy is a highly sensitive technique used for measuring the optical extinction coefficient of gases and particulate matter (O'Keefe and Deacon, 1988; Romanini et al., 1997) without the need for instrument calibration." I'm not sure it's 100% correct to say that CRDS does not require calibration (e.g. Toole et al., 2013). The raw CRDS signal also needs to be adjusted to take into account RL, which can require calibration (see below).

This is a good point and a nice study by Toole et al. (2013), which eliminates some uncertainty in DMA size-selected diameters and CPC uncertainties by effectively calibrating their CRDS. Hence, we have reworded the sentence in the manuscript (P5L10-11) such that it now reads:

"Cavity ring-down spectroscopy is a highly sensitive technique used for measuring the optical extinction coefficient of gases and particulate matter (O'Keefe and Deacon, 1988; Romanini et al., 1997)."

Please see comment 9 for details regarding determination of the CRDS  $R_L$  factor.

7. P4L32: Please provide ring-down time constants for the two CRDS channels.

We have added the following to the manuscript (P5L27-30):

"The  $\tau_0$  times for both the 405 and 658 nm CRDS channels used in this study were measured before and after experiments where aerosol was passed through the optical cavities. These  $\tau_0$  varied over time by only a small amount due to changes in cavity alignment, cleanliness and the sample pressure. However, typical representative times were 23.1 µs (405 nm) and 34.2 µs (658 nm)."

8. P5L3-4: What is the radius of curvature of the CRDS mirrors?

We have modified the following in the manuscript (P5L17-19):

"Cavity mirrors were manufactured from fused silica with wavelength-specific coatings, 25 mm diameter, 1m radii of curvature and reflectivities in excess of 99.99 % (Layertec GmbH, red 660 nm; CVI Laser Optics, blue 405 nm)."

9. P5L12-13: "Cavity mirror-to-mirror lengths ranged from 371-423 mm yielding RL factors in the range 1.150-1.173." How were the RL values measured? While determining RL using the physical dimensions of the CRDS cell may be appropriate for aerosol particles (Langridge et al., 2011), Fuchs et al. (2008) found that for gases RL was not equal to the geometric RL. Were any experiments performed to determine if RL in

your system is different for gas and particles? Is there a reason for the different cavity lengths?

We have modified the following sentence (P5L30-31):

"Cavity mirror-to-mirror lengths ranged from 371–423 mm yielding geometric  $R_L$  factors in the range 1.150–1.173."

The following was also added to the manuscript (P5L31-P6L7):

"The  $R_L$  factor appropriate for aerosol measurements was determined from the geometric dimensions of the detection cell. As highlighted by Fuchs et al. (2008), the  $R_L$  factor for detection of gaseous species can be different from this value, due to the ability of gaseous samples to diffuse. We determined the gaseous  $R_L$  factors by measuring the change in the ring-down times for filtered air plus ozone in (i) standard operation whereby ozone partially diffuses into the volume between the sample inlet and mirror and (ii) non-standard operation whereby ozone was fully mixed into the volume between the sample inlet and mirror by pulling the ozone-laden air out of the cavity through the mirror purge lines. This resulted in  $R_L$  factors 1.05 (658 nm) and 1.04 (405 nm)."

Whilst propagating the ozone  $R_L$  factors through the calibration procedure did not impact on the overall result of this study, the mean gradients in Figure 7 changed from 0.98 ± 0.01 to 1.08 ± 0.01. Figure 7 has been updated accordingly in the manuscript. Also, the following has been modified in the manuscript (P15L12-15):

"The mean gradient between the modelled and PAS-measured absorption cross sections for nigrosin for all five ozone-calibrated PAS cells was 1.08  $\pm$  0.01 (2 $\sigma$  fitting uncertainty) as shown in Fig 7. Gradients for the 405, 514 and 658 nm wavelengths were 1.08, 1.07 and 1.09 respectively."

The following has been modified in the manuscript (P17L4-6):

"Using nigrosin aerosol with mobility-selected diameters in the range 250–425 nm, we verified that the measured absorption cross sections using photoacoustic spectroscopy agreed with modelled values to within 8 %. "Our result is robust for the optical wavelengths 405, 514 and 658 nm."

The different cavity lengths are due to physical size constraints.

10. P5L14-15: See comment 5.

We have reworded the following in the manuscript (P6L8-10) so that it now

reads:

"Extinction coefficients were converted to ensemble extinction cross sections (m2) by dividing by the aerosol number concentrations measured using a CPC (see Sect. 2.4). The ensemble extinction cross section is hereafter referred to as the extinction cross section."

11. P5L20: Were the CRDS cells made of teflon or metal? If they were metal, please specify the material.

We have added the following sentence to the manuscript (P5L17-18):

"The CRDS cells were made of aluminium."

12. P6 Figure 1: This figure was hard to understand at first because I expected the colors of a given box to correspond to the wavelength of that instrument. I think the figure would be clearer if the colors of the PAS and CRDS cells corresponded to the wavelength used for that cell. Perhaps then use different shapes to differentiate between the CRDS and PAS cells?

This is a good suggestion. Figure 1 has been modified in the manuscript.

13. P6L7: "... the measured ozone concentrations were used directly." How were the ozone concentrations measured (see comment below), or do you mean measured extinctions were used directly?

We have reworded the following in the manuscript (P7L7-11):

"For PAS cells in series with the CRDS channels (PAS 4 and PAS 5), the CRDSmeasured extinction coefficients were used directly to calibrate the corresponding in-line PAS channel measurements of IA. This calibration relation between sample extinction and IA is quantified at multiple values of ozone concentration, controlled by varying the discharge frequency on the coronal ozone generator."

14. P6L13-14: "At the start of each calibration cycle, pure oxygen was introduced into the PAS cells through the ozone manifold. The oxygen displaced a fraction of the filtered-air flow through each cell. . . " This sentence and Eq. 2 imply that the flow through the system was a mix of air and gas from the ozone generator. Is this correct? If so, what fraction of the flow came from the ozone generator? How was the ambient air filtered?

The flow was a mixture of pure oxygen (the ozone generator was not powered at

this point) at a flow rate 0.02 L min-1 and filtered air at a flow rate of 0.98 L min-1. The filtered air flow was filtered using a particle-filter. Hence the following has been modified in the manuscript (P8L1-4):

"At the start of each calibration cycle, pure oxygen was introduced into the PAS cells through the ozone manifold at a flow rate of 0.02 L min-1 per cell, in addition to the 0.98 L min-1 filtered-air flow. Air was filtered using a particle filter (Headline Filters, DIF-LK40). The oxygen displaced a fraction of the filtered-air flow through each cell, changing the gas composition, speed of sound and thus cell resonant frequency, as shown in Fig. 2."

15. P6L20: "The 515 nm PAS cell..." Isn't the wavelength 514 nm?

Yes, the wavelength is 514 nm. We have reworded the following sentence in the manuscript (P8L9-10):

"The 514 nm PAS cell was calibrated using the 658 nm CRDS cell, and hence the ozone splitting ratio between PAS cells 3 and 4 was used."

16. P6L21-22: "Ozone splitting ratios derived using this method compared extremely well to in-line mass flow measurements and were in the range 2-28%." What do you mean by "2-28%." Do you mean that the difference in the flow between two cells was between 2 and 28%, or that the splitting ratio,  $\Delta v$ , calculated from Eq. 2 was between 2 and 28%?

To clarify this point, lines P8L10-14 has been changed to:

"The ozone splitting ratio represents the fractional difference in the ozone concentrations within two PAS cells due to unequal flow splitting within the ozone manifold. The ozone splitting ratios, and therefore the ozone-laden flow rates, between two PAS cells located in parallel (for example, the PAS 2 and PAS 4 cells) were in the range 2–28 %. Measuring the ozone splitting between PAS cells using the resonant shift method compared extremely well to in-line mass flow measurements."

17. P7L7-9 and P8 Figure 3: "Ozone concentrations in the range ~10-500 ppm were used." How did you determine the ozone concentrations? Figure 3 shows a maximum extinction of 27 Mm-1 at 405 nm. If this is only from ozone, this gives an ozone concentration of ~660-750 ppmv ( $\sigma_{ozone}$  around 405 nm is 1.45 – 1.65 × 10-23 cm2, depending on the exact wavelength (Serdyuchenko et al., 2014)), higher than the 500 ppmv in the text. Also, 10 ppmv of ozone gives an extinction of ~50 Mm-1 at 658 nm. Did you put lower ozone concentrations into the green and red

PAS cells to extend the calibration curves to lower values? Were the same ozone levels used for both the 405 and 658 CRDS channels? If so, how do the ozone concentrations calculated using the measured extinction and the literature cross sections compare for those two wavelengths?

We have updated the precision of the ozone concentrations, calculating them by dividing the extinction coefficient by the ozone absorption cross section at a wavelength of 405.03 nm and assuming  $2.46 \times 10^{25}$  molecules of air per cubic metre at the 405 nm CRDS cell temperature and pressure of 21.82 °C and 1001 mb. The following paragraph in the manuscript has been modified (P8L22-P911):

"Calibrations involved the stepwise measurement of nine ozone concentration levels, where Fig. 3 shows the PAS and CRDS responses to ozone at 405 nm. Using the minimum and maximum extinction coefficients for ozone in Fig. 3 (1.3 and 27.1 Mm-1, respectively), an ozone absorption cross section of  $1.62 \times 10^{-23}$  cm2 at the corresponding CRDS wavelength (405.03 nm) and assuming  $2.46 \times 10^{25}$  molecules of air per cubic metre at the 405 nm CRDS cell temperature and pressure of 21.82 °C and 1001 mb, the ozone concentrations were in the range 33–680 ppmv (Serdyuchenko et al., 2014). Approximately the same levels of ozone were used in all cells. The ratios of ozone extinction coefficients measured in the 405 and 658 nm CRDS cells compared well to the ratio of the literature ozone absorption cross sections. After accounting for uneven ozone flow splitting between the cells, the ratio of the measured extinction coefficients at 658 and 405 nm agreed with the literature cross section ratio to within 2.0 %. This excellent agreement provides strong evidence that there were no issues with contamination by absorbing gaseous or aerosol species during ozone calibrations."

18. P7L17 (Eq. 3): Why is the resonant frequency represented by  $\nu$  in this equation, and  $F_{r}$  in Eq. 2?

The following line in the manuscript has been modified (P9L18, Eq. 3) so that  $\nu$  has been replaced with  $F_{R}\!.$

19. P7L18: How is PL measured? By the photodiode? Also, what are typical quality factors for your instrument?

The following lines have been added to the manuscript (P9L19-20):

*"PL* was measured by the photodiode. PAS cell quality factors were in the range 87– 93."

20. P8L1 (and elsewhere): I might consider replacing "gradient" with the

more common "slope," but this is mostly preference on my part.

We would prefer to maintain our original wording.

21. P8 Figure 3: The y-axis units are inverse megameters  $(Mm^{-1})$ , while the units in the text (e.g. P4L23 and P5L9) are inverse meters  $(m^{-1})$ . Since  $Mm^{-1}$  are the customary units in aerosol work, I would suggest changing the units in the text, but whatever you choose, the units should be consistent.

This is a good point. We have changed all units of absorption and extinction coefficients to Mm-1.

22. P8L10: Why was the red CRDS used to calibrate the green PAS? Do you get the same result if you use the blue CRDS instead?

We have added the following to the manuscript (P10L10-14):

"The 658 nm CRDS was used to calibrate the 514 nm PAS channel because it extended over a greater range of extinction coefficients (167–1506  $Mm^{-1}$ ) than the 405 nm CRDS (1–27  $Mm^{-1}$ ). This ensured that the 514 nm PAS calibration covered a range of absorption coefficients greater than that spanned by the nigrosin absorption coefficients. Calibrating the 405 nm channel using the 405 nm CRDS channel, as opposed to the 658 nm channel, would lead to absorption coefficients that were lower by 3.2 %."

23. P8L17: Please provide the product number and lot number for the nigrosin used.

We have modified the following in the manuscript (P10L21):

"Water-soluble nigrosin, a strong light-absorbing dye at visible wavelengths, (Sigma Aldrich, CAS Number 8005-03-6, lot number MKBR1705V, product number 198285-100G) was dissolved into high purity deionised water (VWR Chemicals) with a range of concentrations between 3.2-7.1 grams per litre (g L-1)"

24. P9L2: What was the DMA sheath flow?

We have added the following in the manuscript (P11L6-8):

"Flow rates through the mass flow controller were set to regulate the flow through the DMA such that the sample-to-sheath flow ratio was at least 1:10 with a sample flow rate in the range 0.3-0.4 L min-1 and sheath flow rate in the range 3.5-4.0 L min-1."

25. P9L11: "The aerosol flow was split between optical cells using a series of Y-flow splitters." Please show how this was done in Figure 1.

We have adjusted Figure 1 (P9L11).

26. P12 Figure 5e-f: When I use either the Bohren and Huffman Mie codes or an online Mie calculator (https://omlc.org/calc/mie\_calc.html) to calculate absorption cross sections, I get numbers lower than those shown in Figure 5e. For example, for 1000 nm diameter particles, I get the following absorption cross sections:  $1.01 \times 10^{-12}$  m2 at 405 nm;  $1.06 \times 10^{-12}$  m2 at 514 nm; and  $1.10 \times 10^{-12}$  m2 at 658 nm. The absorption cross section that I calculate for 400 nm particles at a wavelength of 514 nm ( $1.69 \times 10^{-13}$  m2) is higher than the value shown in Figure 5f ( $\sim 1.4 \times 10^{-13}$  m2) Please explain these discrepancies. If there was an error in the Mie calculation, how does this affect the agreement between the measured and modeled aerosol extinction and absorption cross sections?

Our apologies, this was due to a plotting inaccuracy where the figure was incorrectly modified to display the absorption cross section, which has now been amended in the manuscript. All other instances in the analysis script are correct and the result of the paper is not impacted.

27. P12L19-20: See comment 5.

We have clarified this point by modifying the following sentence in the manuscript (P15L5-7):

"The ensemble extinction cross sections (hereafter referred to as extinction cross section) for nigrosin with mobility-selected diameters in the range 250–425 nm were measured using CRDS and modelled using Mie theory, as outlined in Sect. 2.5."

28. P13 Figure 6 and P14 Figure 7: Are the measured cross sections effective cross sections (measured extinction or absorption divided by the particle concentration) or are they corrected for the effect of multiply charged particles?

They are ensemble cross sections. The labels in Figures 6 and 7 have been modified (P15-16).

29. Supplementary material, Tables S1 and S2: What do "gdry," "rtd," "btd," and the other abbreviations in the second column of Table S2 stand for? I'm assuming that these are the names of the different CRD/PAS cells. If so, the names in the text, in Figure 1, and in the supplement should all be consistent.

We have modified the labels in tables S1 and S2 to be consistent with the rest of the manuscript.

30. Supplementary material, Table S2: I'm assuming that "bdry" and "btd" refer to the two 405 PAS cells. If so, why is the gradient for the bdry/bdry PAS/CRDS calibration 25% higher than the gradient for the btd/bdry PAS/CRDS calibration (and the same for rtd/rdry and rdry/rdry)? If you put the blue CRDS in front of PAS 1 instead of PAS 5, do you get the same gradients?

In the first instance 'bdry/bdry' refers to one of the 405 nm PAS cells and the 405 nm CRDS cell and the second instance 'btd/bdry' refers to another 405 nm PAS cell and the same 405 nm CRDS cell. One reason for the difference between the gradients for the two PAS cells is due to different microphone sensitivities. However, it is not clear what the reviewer is referring to by 25 % differences in the gradient.

**Technical corrections**

All the 'et al.' instances have been addressed. The 'M' in 'McManus' has been capitalised in both instances. All missing doi have been added where appropriate.